# Immune Responses to Some Viral Infections That Have a High Evolutionary Potential—A Case Report with Literature Review

**DOI:** 10.3390/life12070940

**Published:** 2022-06-23

**Authors:** Maria Lucia Sur, Bogdan-Stefan Moldovan, Diana Mocanu, Gabriel Samasca, Iulia Lupan, Ionel Armat, Marin Harabagiu, Genel Sur, Calin Lazar

**Affiliations:** 1Department of Pediatric I, Iuliu Hatieganu University of Medicine and Pharmacy, 400006 Cluj-Napoca, Romania; sur.maria@umfcluj.ro (M.L.S.); marinharabagiu@gmail.com (M.H.); calinlazar@umfcluj.ro (C.L.); 2Children’s Emergency Clinical Hospital, 400370 Cluj-Napoca, Romania; bogdanmoldovan1106@gmail.com (B.-S.M.); dianna.mocanu@yahoo.com (D.M.); armat_95ionel@yahoo.com (I.A.); surgenel@yahoo.com (G.S.); 3Department of Immunology, Iuliu Hatieganu University of Medicine and Pharmacy, 400006 Cluj-Napoca, Romania; 4Molecular Biology Department, Babes Bolyai University, 400084 Cluj-Napoca, Romania; iulia.lupan@ubbcluj.ro; 5Clinical Hospital of Infectious Diseases, 400000 Cluj-Napoca, Romania

**Keywords:** viral infections, infectious associations, treatment

## Abstract

Viral infections are a key issue in modern medicine. SARS-CoV-2 infection confirms that we are not sufficiently prepared for these unforeseen infections. The COVID-19 pandemic has cultivated a great sense of fear and distrust in patients. If viral infections, in this case, SARS-CoV-2, overlap with another infection, the symptoms are prolonged and worsened, and complications may occur. Starting from an objective clinical finding of a patient they had in follow-up and treatment, the authors present the problems of the diseases the patient suffered from. These are described as reviews so that readers can get an idea of the clinical methods of expression and the therapeutic possibilities. Therefore, this article describes Lyme disease and post-treatment Lyme disease syndrome, SARS-CoV-2 infection, and multisystem inflammatory syndrome in children (MISC-C), as the patient suffered from an incomplete form of Kawasaki disease. During the treatment for Lyme disease, the patient also contracted the influenza type A virus. Although any of these diseases could have the potential for serious evolution, our patient still went through these infections relatively well. This can be explained by the fact that the patient had a slow immune response to the aforementioned infections, which allowed him to survive these diseases relatively easily, unlike other individuals who have an exaggerated immune response or who suffer from serious immune involvement, e.g., hepatitis B with a fulminant response. The case was presented chronologically, but at the same time, all particular infection manifestations were accurately described. For these reasons, the article is presented in the form of a review, exemplified by the case itself. Of the 52 cases of MISC-C found in the Pediatrics Clinic II of Cluj-Napoca, we present the case of a male patient who presented with Lyme disease, post-treatment Lyme disease syndrome, Kawasaki disease, and MISC-C incomplete form.

## 1. Introduction

Lyme disease is a multisystemic disease caused by infection with Borrelia burgdorferi. It is also known as the “disease with a thousand faces” because the symptoms can mimic various other diseases. It is characterized by a variety of acute symptoms but also late manifestations affecting various organs and systems such as the skin, musculoskeletal system, central nervous system, and heart [1]. Some patients with Lyme disease have symptoms after months or years of antibiotic treatment, which is known as post-treatment Lyme disease syndrome (PTLDS). From the etiopathogenetic point of view, the syndrome is unclear and its management is not very well defined [2]. Lyme disease is common in Europe, North America, and Asia, and it affects 5–10% of tick-borne patients [3,4]. In epidemic regions, the incidence of the disease is 50–150/100,000 individuals. A German study of 12,000 patients found a prevalence of 1.3 in children aged 1–2 years old and 7.1 in adolescents aged 14–17 years old [4]. Clinically, Lyme disease may have localized or disseminated symptoms with early or late manifestations. The most common manifestation of the disease is migratory erythema, which is similar to maculopapular erythematous lesion with a clear center surrounded by circular erythema that lasts several days and appears at the site of the bite, 1–3 weeks after the bite. Skin manifestations are present in 70% to 90% of cases. Other symptoms are present in the central nervous system, skeletal muscles, and heart, which are described in Table 1 [3,5,6,7,8].

## 2. Post-Treatment Lyme Disease Syndrome

Months after treatment, many patients report persistent symptoms such as fatigue, myalgia, arthralgia, and neurocognitive dysfunction. The majority of symptoms in this syndrome are associated with the joints and the central nervous system [6,9,10,11,12,13,14]. In 2005, the Swedish Society of Infectious Diseases and, in 2006, the American Society of Infectious Diseases tried to define this syndrome. This syndrome is considered complex with nonspecific and subjective symptoms such as fatigue, musculoskeletal pain, and neurocognitive dysfunction. This syndrome is considered a continuation of the disease and is not produced by other clinical entities. These societies showed that 5% to 10% of patients experience persistent symptoms lasting more than 6–12 months after antibiotic treatment for Lyme disease [6]. Many studies have not shown a difference between the general population and this syndrome after antibiotic treatment in terms of symptoms. Some authors believe that the disease is being exacerbated by immune mechanisms and that the presence of the bacterium B.burgdorferi cannot be proven. Most authors do not recommend treatment with antibiotics. However, it is still considered that ceftriaxone (Cefort) would have positive effects on the symptoms [15].

## 3. SARS-CoV-2 Infection

Since the beginning of the pandemic, SARS-CoV-2 infection has affected a smaller proportion of the pediatric population and there have only been mild cases. The number of cases has increased significantly in children since the onset of the delta and omicron mutations, but the number of serious cases has been low [16]. It has been found that some pediatric patients may develop a life-threatening condition called multisystem inflammatory syndrome in children (MISC-C). This MISC-C, in most cases, can be considered a post-infectious manifestation secondary to an abnormal immune response. It occurs in previously healthy patients [17]. If the manifestations of this syndrome are severe, the cases should be hospitalized in a clinical or even intensive care unit. Impaired vital signs include vital signs (tachycardia, tachypnea), shock, respiratory failure, heart damage, similarities to Kawasaki disease, neurological changes with depressed mental status, changes in neurological examination, seizures, severe abdominal pain or vomiting, evidence of renal and hepatic impairment, and coagulation disorders [18]. The clinical picture of this syndrome can evolve in two directions: in young children with symptoms similar to Kawasaki disease, toxic shock syndrome, or macrophage activation syndrome, and in older children, it predominates in septic shock syndrome [16,17,19]. The onset of manifestations occurs 2 to 6 weeks after acute infection with the SARS-CoV-2 virus; in a few cases, it has been found to appear after 6 weeks after the infection [20]. The symptoms and signs are diverse, and patients who have mild forms at first presentation may develop severe forms of the disease within a few days [21,22]. Important and critical signs are left heart failure, hemodynamic instability, and respiratory distress [23,24].

Fever is an important element that can be present for up to 6 days. Fever is a mandatory symptom of MISC-C syndrome, and its presence for at least 3 days raises suspicion of the disease [25,26]. Heart damage is a clinically important manifestation that may occur in the early or late stages of evolution. It is characterized by myocardial dysfunction, valvular dysfunction, pericarditis, atrioventricular block and other arrhythmias, and hypotension with shock may occur. It is very important to look for coronary dilations that can range from small to giant aneurysms. These dilations may occur a few days after the disappearance of MISC-C symptoms [17,27,28]. Respiratory impairment is characterized by tachypnea, dyspnea, and less frequent coughing. Symptoms may be limited to the upper respiratory tract or may present as a form of acute pneumonia [29]. Digestive impairment is common and is characterized by abdominal pain, nausea, vomiting, and diarrhea [30,31]. Muscle weakness and reduced reflexes may occur, but central nervous system manifestations may also occur with headaches, confusion, lethargy, irritability, dysarthria, dysphagia, meningism, and cerebellar ataxia. In a few cases, severe encephalopathy has also occurred in aseptic encephalitis or meningitis, as well as a demyelinating disease of the central nervous system (Guillain–Barre Syndrome) [32,33]. Skin-mucosal damage is characterized by a transient rash with swollen and red lips, a raspy tongue, and swelling of the hands and feet with scaling. It was found that in those with skin manifestations, the level of inflammatory markers is lower and the frequency of shock is lower [34]. Renal impairment with acute renal failure occurs in over 38% of cases. However, the need for renal dialysis predominates in less than 0.1% of them [26].

In general, at least four organs are involved in MISC-C, but there are cases involving several organs [17]. A meta-analysis that included 17 studies with 992 patients with MISC showed that fever was present in 95% of cases, gastrointestinal signs and symptoms in 78%, cardiovascular in 75%, and respiratory in 55% [16]. In a study conducted in the Second Pediatric Clinic of Cluj-Napoca, 52 cases of MISC-C were followed. It was found that in young patients, the symptoms of Kawasaki-like predominate, and in older ones, the symptoms of organ failure predominate.

Of the 52 cases of MISC-C found in the Pediatrics Clinic II of Cluj-Napoca, we present the case of a male patient who presented with Lyme disease, post-treatment Lyme disease syndrome, Kawasaki disease, and MISC-C incomplete form.

On 20 August 2021, the patient was bitten by a tick at the right retroauricular level. Initially, the tick was removed by his mother, but the tick’s head remained stuck in the skin, which is why he presented to the Emergency Department of Pediatrics in Cluj-Napoca. The doctor recommended symptomatic treatment at home. After about 1 week, the patient developed an erythematous maculopapular rash with a blister in the center at the retro auricular level in the bite area (Figure 1) and altered general conditions, including fever, headache, nausea, and arthralgia. He went a second time to the Pediatric Emergency Department, where the doctor recommended antibiotic treatment with Cefuroxime 500 mg/day, once a day for 21 days.

The patient’s mother decided, after 3 days, to consult an infectious disease doctor at the Clinical Hospital of Infectious Diseases Cluj Napoca, who clinically established the diagnosis of Lyme disease and recommended treatment with Cefuroxime 750 mg/day, once a day for 21 days. Laboratory determinations were performed in order to demonstrate the existence of Lyme disease in the presence of suggestive clinical manifestations of marginal erythema, fever, altered general condition, and arthralgias. Analyses with the Elisa Mikrogen Diagnostik Kit revealed Ig M positivity for Borellia burgdorferi, as well as Western Blot IgM positivity and IgG negativity. The antibodies for hepatitis C virus and Mycoplasma spp. were non-reactive, and the rheumatoid factor was within the range limit.

After 21 days of correct treatment with Cefuroxime, the symptoms persisted for more than 6 months after the end of antibiotic therapy, with recurrent headaches, arthralgia, dizziness, and cognitive impairment. It was interpreted as a post-treatment Lyme disease syndrome (PTLD). The teacher also realized that the patient no longer had the same memory strength. During this time, he intermittently took symptomatic treatment such as nonsteroidal anti-inflammatory drugs for headaches and arthralgia. It should be noted that these symptoms were present daily according to the child’s words. In January, the patient was diagnosed with a SARS-CoV-2 infection and presented with a mild form of fever, altered general condition, headache, and arthralgia, and remained isolated at home for 14 days. Because the symptoms were not severe, he received only symptomatic treatment.

Approximately 3 weeks after infection with the SARS-CoV-2 virus, the patient had a fever for 3 days, redness of the lips, erythema of the palms (Figure 2) and soles but no flaking, a fleeting erythematous rash on the body, and small palpable laterocervical lymphadenopathy, associated with abdominal pain and loss of appetite.

Evidence of inflammation was also present, including increased D-dimers, neutrophilia, and lymphopenia. He took ibuprofen and supportive treatment at home, with relatively good evolution. It should be noted that the initial symptoms of headache, arthralgia, and myalgia remained, and he was admitted to our clinic in January, where blood tests were performed. It showed that the full blood count, liver and kidney tests were normal, also without the inflammatory syndrome, creatine phosphokinase (CPK) was within normal limits, and anti-streptolysin O antibodies (ASLO) were within normal limits, so a post-streptococcal pathology is excluded. IgA, IgG, and IgM levels were also within normal limits with the exclusion of immunodeficiency, and rheumatoid factor and rapid plasma reagin (RPR) were normal. Negative were antinuclear antibodies (Ac ANA), anti-double-chain antibodies (Ac DS-DNA), and anti-thyroperoxidase (anti-TPO) antibodies were negative. A neurological examination was also performed with an electroencephalogram (EEG), but this did not detect any clinically significant changes. It was recommended by the pediatric neurologist to perform a brain MRI where no pathological changes were detected. The patient’s symptoms persisted and he was hospitalized again in our clinic in February, so we decided to introduce antibiotic treatment with Ceftriaxone 80 mg/kg/day for 21 days with the support of the digestive bacterial flora (probiotics) and anti-inflammatory drugs. The evolution was good; the child rarely complained of headaches and arthralgias. After 10 days of treatment, the patient had a fever (40°), chills, muscle aches, joint pain, and altered general condition. The patient followed a symptomatic treatment, and after 3 days of evolution, the symptoms improved. Laboratory tests showed the presence of the influenza A virus.

The main problem that caught our attention, in this case, was the particularity of the immune response. He had a slow response, which did not lead to serious side consequences. We know that there is a possibility of a strong response in which the affected organs also suffer, an example being the destruction of hepatocytes by the immune system in viral hepatitis B with a strong response. Our case did not have the full form of MIS-C and no serious form of influenza A, which showed us once again the slow but sufficient response to recover from these infections.

The short-term prognosis of the case was favorable, with the patient being able to successfully recover from the described infections. We consider that the long-term prognosis remains good. The symptoms that the child described disappeared, and now he has a normal social status and good results at school.

## 4. Diagnostic Possibilities in SARS-CoV-2 Infection

### 4.1. Laboratory Investigations

In all cases, the SARS-COV-2 RT-PCR test and serological tests must be performed upon hospitalization [35,36]. Investigations for MISC-C syndrome will be carried out in stages, depending on the symptoms:

#### 4.1.1. Level I

Complete blood count, ESR, CRP, urine tests, electrolytes, and albumin level. If the levels of PCR and ESR are elevated and lymphopenia, thrombocytopenia, hypoalbuminemia, hyponatremia, and neutrophilia are associated, then move to level II of investigations [29,37,38,39].

#### 4.1.2. Level II

Ferritin, LDH, procalcitonin, blood glucose, electrolytes, arterial blood gases (ABG), liver enzymes, urea, creatinine, cardiac markers: NTpro-BNP, Troponin and will also evaluate triglycerides, prothrombin, D-dimers, fibrinogen, and for the differential diagnosis of hemophagocytic lymphohistiocytosis (HLH) or other malignant haemopathies will perform bone marrow examination [40,41].

Bacteriological and viral investigations will be conducted for the differential diagnosis [42].

Medium and severe forms will include uroculture, coproculture, pharyngeal or nasal secretions, and cerebrospinal fluid analysis. Serology for leptospirosis, Lyme disease, and mycoplasma pneumonia can also be performed. By performing the Quantiferon test, tuberculosis is excluded, and the human immunodeficiency virus (HIV) is also excluded. Virological investigations are also needed to exclude Epstein–Barr virus (EBV), cytomegalovirus (CMV), enterovirus, and adenovirus [18]. All patients with suspected MISC-C should have an inpatient ECG and echocardiography to assess cardiac function and coronary artery dilation.

#### 4.1.3. Positive Diagnosis

All patients who meet the WHO/CDC criteria are considered MISC-C. An important note is that all patients who have complete or partial criteria for Kawasaki disease should be considered for MISC-C. The WHO proposes to meet the six criteria for classification [43]:Children and adolescents aged 0 to 19 years old.Fever lasting more than 3 days.Two of the following: skin manifestations similar to Kawasaki disease, hypotension or shock, signs of myocardial, pericardial, or valvular dysfunction, coronary abnormalities as well as changes in troponin and NT-proBNP levels, changes in coagulopathy (prothrombin PT and the partial thromboplastin time APPT as well as D-dimers), acute gastrointestinal symptoms: vomiting, diarrhea, and abdominal pain.Inflammation markers increased: ESR, CRP, and procalcitonin (PCT).Elimination of obvious causes of bacterial infection: bacterial sepsis, staphylococcal or streptococcal toxic shock syndromes.RT-PCR SARS-COV-2 positive, SARS-COV-2 antigen-positive, serology positive, or contact with a person positive for SARS-COV-2 virus.

#### 4.1.4. Differential Diagnosis

According to the American Academy of Pediatrics, any child suspected of having MISC-C should be evaluated for infectious and non-infectious etiologies. Differentiation between MISC-C and Kawasaki disease is presented in Table 2.

Studies have shown that between 40% and 50% of children with MISC-C meet the criteria for complete or incomplete Kawasaki disease. The incomplete form of Kawasaki disease is defined by persistent fever and fewer than four of the classic symptoms, with suggestive laboratory and echocardiographic data [44,45].

Bacterial infections may be common with MISC-C, but bacterial infections involve a single organ or system, and MISC-C involves multisystemic organs. Severe infections with fever, rash, and shock may be caused by leptospirosis, rickets, and exposure to animals, ticks, and mosquitoes. Toxic shock syndrome (TSS) is caused by Streptococcus spp. and Staphylococcus spp. and has similar characteristics to MISC-C. In the case of viral infections, a similar concordance with MISC-C is rare. The EBV virus can affect the central nervous system, liver, lungs, and heart and can trigger secondary haemophagocytic lymphocytosis (HLH) [46]. HLH and macrophage activation syndrome (MAS) have some characteristics similar to MISC. Systemic lupus erythematosus (SLE), which can cause serious damage to the kidneys and central nervous system, is not characteristic of MISC.

## 5. Treatment of MISC-C

Unfortunately, there are currently no studies comparing the effectiveness of different treatment options. Treatment will be based on the severity of the symptoms. Mild forms require only monitoring without corticosteroids or immunoglobulins.

### 5.1. Immunomodulatory Anti-Inflammatory Treatment

In the first stage of treatment, if there is shock or severe damage to one or more organs, immunoglobulins plus methylprednisolone are recommended. Immunoglobulins will be administered intravenously at 2 g/kg/dose in a single dose, with a maximum of 70 g in a single infusion for 12 h or 16 h in those with heart failure. For those with heart disease, immunoglobulins may be given at 1 g/kg/day for 2 days in a row. Methylprednisolone is provided intravenously in 1.6–2 mg/kg doses divided into two doses for 5 days [47] after which it is switched to oral therapy and prednisone 1–2 mg/kg with a dose reduction in the next 3–4 weeks. Depending on the evolution of the symptoms, the cortisone treatment can be administered for 2–3 weeks or even longer. [36] In some refractory patients, higher doses of methylprednisolone (10–30 mg/kg/day) may be given as an infusion of up to 1 g/day for 1–3 consecutive days, followed by oral prednisone of 1–2 mg/kg/day for 7 days and a dose reduction within 2–3 weeks [47,48].

In stage II of treatment, in refractory cases to the treatment in stage I or in the case of the combination of HLH, Anakinra, an IL1 receptor antagonist, at a dose of more than 4 mg/kg/day, intravenous or subcutaneous administration is recommended [48]. Patients with MISC-C require long-term immunomodulatory treatment to avoid recurrence of hyperinflammatory syndrome, which should be given for at least 2–3 weeks.

Tocilizumab, an IL-6 receptor antagonist, is recommended only in cases refractory to immunoglobulins and methylprednisolone or if Anakinra is not available. The recommended dose is 8 mg/kg in a single dose for those under 30 kg and 12 mg/kg for those over 30 kg [49].

### 5.2. Antiviral Treatment

Remdesivir is recommended for patients who have a positive RT-PCR SARS-CoV-2 test. The dose of Remdesivir is 5 mg/kg intravenous on the first day and then 2 mg/kg intravenous for the next 9 days [18].

### 5.3. Anticoagulant and Antiplatelet Agents

Anticoagulants are recommended for patients with heart damage or coronary dilatation. In patients with documented thrombosis or coronary artery aneurysm, Enoxaparin should be administered every 12 h in children less than 2 months, at 1.5 mg/kg/dose, and in children over 2 months, at 1 mg/kg/dose [50].

Patients with moderate or severe MISC-C or Kawasaki-like symptoms [18] should receive aspirin in low doses (3–5 mg/kg/day), while monitoring for Reye’s syndrome. Patients with documented thrombosis should continue anticoagulant therapy for at least 3 months after discharge.

### 5.4. Antibiotic Treatment

Because patients with MISC-C may initially show signs of septic shock, empirical antibiotic therapy should be given initially. The first intention is to recommend cefotaxime or ceftriaxone with vancomycin. Piperacillin-tazobactam may also be used for digestive infections [47]. In cases of severe shock, vancomycin, clindamycin, and cefepime or vancomycin, meropenem, and gentamicin [51] or cefotaxime, daptomycin, and clindamycin are recommended [47].

## 6. Conclusions

Our subject’s particular immune response, represented by a slow reaction to viral infections, makes him survive these infections. Our case involves some viral infections, each of which could have caused serious health problems. However, due to this slow and modular immune response, he managed to survive.

## Figures and Tables

**Figure 1 life-12-00940-f001:**
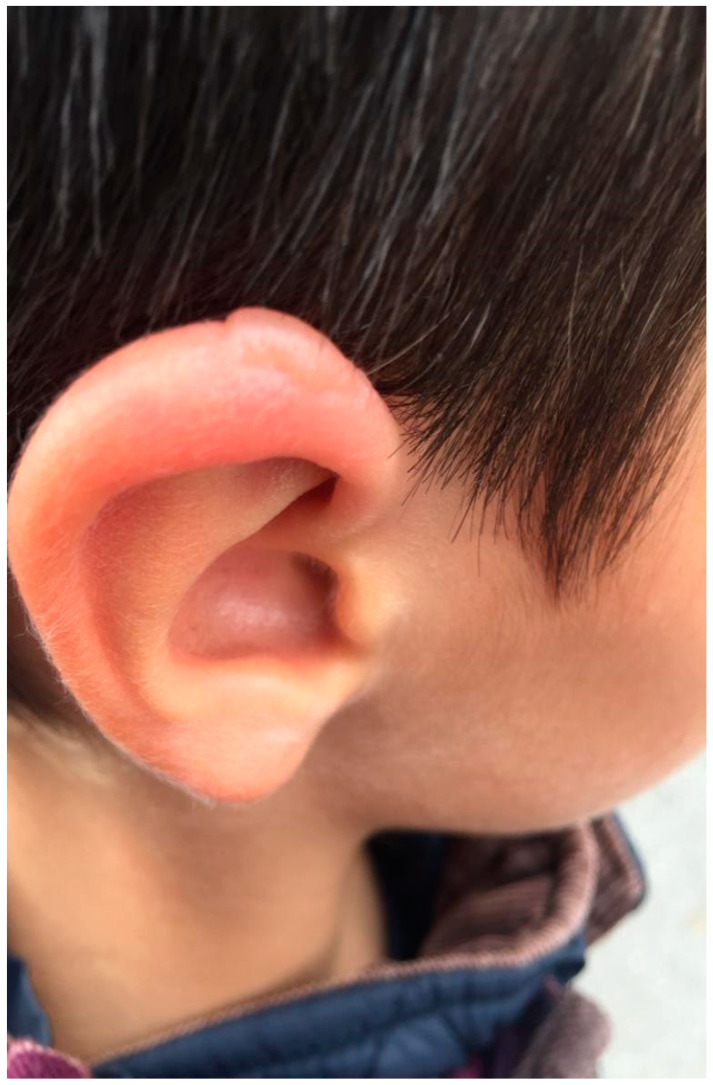
Acute erythema migrans in Lyme disease.

**Figure 2 life-12-00940-f002:**
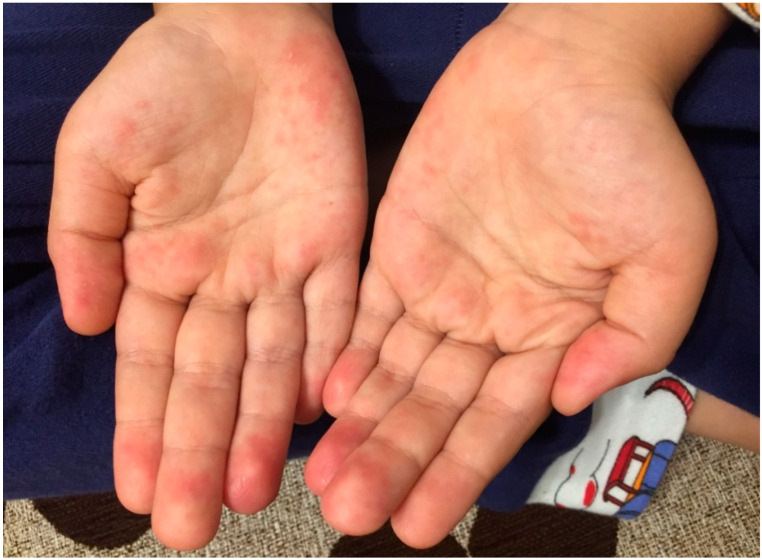
Palmar erythema in incomplete Kawasaki disease.

**Table 1 life-12-00940-t001:** Clinical manifestations and treatment of Lyme disease.

The Organ System Involved	Early Stage		Late Stage
	Localized	Disseminated	
Skin	**Erythema migrans (EM)**	**Multiple EM** **Lymphocytoma**	**Acrodermatitis chronica atrophicans** (rare in children)
		Dx: clinical Tx: Amoxicillin (<8 years of age) 10–14 days PO or Doxycycline (>9 years of age) 14–30 days PO	
Central nervous system	Not applicable	**Peripheral facial nerve palsy** **Lymphocytic meningitis** **Meningopolyradiculitis** Dx: clinical suspicion + serology in serum and CSF Tx: Ceftriaxone IV 14–21 days OR Doxycycline PO 14–21 days if isolated facial palsy without meningitis and age >9 years	**Chronic encephalomyelitis** Dx: clinical suspicion+ serology in serum and cerebrospinal fluid Tx: Ceftriaxone IV 14–28 days
Musculoskeletal system	Not applicable	**Arthralgia** **Myalgia** Dx/Tx: associated with other manifestations, not established as a standalone manifestation	**Episodic arthritis** **Chronic arthritis** Dx: clinical suspicion +serum anti-B.b Ig G Tx: Amoxicillin (<8 years of age) or Doxycycline (>9 years of age) 28 days PO OR Ceftriaxone IV 14 days
Heart	Not applicable	**AV block** **Myocarditis** **Pericarditis** Dx: clinical suspicion + serology Tx: Ceftriaxone IV 14–21 days OR Doxycycline age >9 years/Amoxicillin age <8 years PO 21 days in outpatients with only first-degree atrioventricular block with PR interval <300 milliseconds	Not applicable

**Table 2 life-12-00940-t002:** The differences between Kawasaki disease and MISC-C.

	Kawasaki Disease	MISC-C
Age	under 5 years	schoolchildren/teenagers
Clinical:		
Fever	+++	+++
Gastrointestinal symptoms	+/−	+++
Myocardical dysfunction	+/−	+++
Structural impairment	+++	+++
Heart:		
Shock	+/−	+++
Left ventricular dysfunction	+/−	+++
Paraclinical:		
D-dimer	+	+++
Ferritin	+	+++
Troponin	+	+++
NT-proBNP	+	+++
CRP	+	++
Lymphopenia	+/−	++
Thrombocytopenia	+/−	++
Thrombocytosis	++	+/−

## Data Availability

Not applicable.

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
