# Peer review of "Immune Responses to Some Viral Infections That Have a High Evolutionary Potential—A Case Report with Literature Review"

_life, 2022, doi:10.3390/life12070940_

Round 1
Reviewer 1 Report
The title is informative and relevant. The references are relevant and recent. The cited sources are referenced correctly. Appropriate and key studies are included.
The introduction reveals what is already known about this topic. The research question is clearly outlined.
The case is well-described, the used methods methods for diagnosing and therapy are valid and reliable.
The patient data is presented in an appropriate way. The illustrative materials are relevant and clearly presented.
Data is discussed from different angles and placed into context without being overinterpreted.
The conclusions are supported by references and own results.
This paper added to what is already in the topic. The article is consistent within itself.
Specific comments on weaknesses of the article and what could be improved:
Major points
- The structure of the paper might be improved. There is no need of Material and methods and results, but Case report description (or something equivalent) and Discussion.
- The title should be changed to mirror the type of paper - it is case report, and also that it is about Lime disease, and not just for many infections. the present title is eye-catching, so, I recommend just to update it by precising the topic of the paper.
- State the prognosis and the follow-up of the patient.
Minor points
- Some minor spelling errors, such as "SARS CoV2" and other variants of the virus name, etc.
- Some additional illustrative materials, such as a figure would be of benefit for the paper (i.e., imaging of the patient, pictures of skin or other, patient journey flow chart, figure on pathogenesis, etc.)
Author Response
Answer to recensor 1
- Regarding the structure of the paper, we have made appropriate changes, in the sense that we have removed the materials, methods, and results part.
- The title has been changed, to reflect better the response to our subject.
- We have made estimations for the long and short-term prognosis in the last part of the paper.
- Drafting mistakes have been corrected.
- We have added additional illustrative materials
Reviewer 2 Report
Dear Authors,
I had the great pleasure to review your work. Please find my comments below:
- The title is very ambiguous, it should be concise, specific and contain key points of the work
- Although you added citations and discussed them widely, I believe is not enough for the article to become a review as in order to cover the topic, the review article should provide a new hypothesis or should concentrate on and summarize a certain point that is needed to be reviewed.
- The abstract should be more structured, e.g., objectives, methods, results, and conclusions. Make conclusions respond to objectives.
- The introduction should be more clear and concise, state the question to be addressed by your study, and why it is being asked. You described clinical manifestations of Lyme disease, Post Lyme disease treatment syndrome, SARS-CoV-2 Infection, and multisystem inflammatory syndrome in children but you should use 1.1, 1.2 when describing more syndromes in the introduction. State the novelty or uniqueness of the research..
- Materials and methods, this section should give enough detail to allow the reader to evaluate the work fully and repeat the experiments exactly, but it should be as brief as possible. Also, you did not describe the study design, the order in which the study was conducted, how the study was conducted, the rationale for the study, ethical approval and consent of the subject.
- The Results section should contain only results pertinent to the question asked in the Introduction. But the section should include results whether or not they support the hypothesis being tested. Organize the results from the most important (usually the question or hypothesis of the study) to the least important. Most data should be presented in tables or figures—and not repeated verbatim in the Results text.
Report statistical analyses. When writing the paper, consider writing this section first; the results are the groundwork for the entire paper. - Begin the discussion with a concise statement of the study’s most important results, especially those that respond (whether positively or negatively) to the question(s) asked initially. Do discuss the significance and importance of your results—without exaggeration or excessive exuberance.
Do compare your results to existing knowledge as indicated. Mention the limitations of your study (only one patient). Suggest future studies that may be indicated on the basis of your findings. Conclude with a clear and concise summary of your results and their significance. Tie the conclusion into your originally stated goals. Make the concluding paragraph or statement(s) agree with those of the abstract. - Tables and figures should be designed clearly and concisely, with informative titles and legends and neatly aligned columns and rows.
Author Response
Answer to recensor 2
- We have modified the title, in the sense that we have included in it the idea of a particular way of immune response to viral infections.
- We think the article corresponds to the review section because it addresses the issue of diseases that our subject has had in detail. At the same time, this case emphasizes only the stager our subject went through. In the abstract, we have added the particular way of a slower immune response, which explains the favorable evolution of the case.
- The abstract explains the particular ways of progress and we do not consider it mandatory to write the materials, methods, and results part.
- The introduction clarifies the difference between a review and a case report due to the new additions we have inserted.
- The materials and methods part has been removed from the study because the manifestations we’ve described were found in only one patient, although we showed that there were 52 cases with MIS-C.
- We have modified the results section, in the form of diagnostic challenges for the diseases that our patient has had.
- To our question about our subject's better tolerability to chain viral infections, the answer is a slow immune response, but enough to rule out the consequences of the disease.
- We consider that the tables are suggestive enough to understand the described mechanisms.
Round 2
Reviewer 1 Report
The paper has been improved significantly. No further issues detected.
Author Response
Dear Reviewer 1
Thank you for your words.
Reviewer 2 Report
Dear Authors,
I appreciate your hard work. Although your article needs more editing I still maintain my previous opinion that it would be more suitable as a case report as you discuss the clinical nature of a single case which illuminates an underlying principle of a disease state, its diagnosis, or its therapy. In my opinion, reviews should give a succinct overview of a particular topic, in your research complications and co-infections of COVID-19.
Author Response
Dear Reviewer 1
1.I believe that the authors did not add much to the previous manuscript.
Changes to the first review were highlighted in red.
2.Although I don`t feel qualified to judge the English language, I find that it needs more editing ( e.g. in the title,, some viral infection").
The title has been redone as suggested. We also worked on English.
3.Also, I maintain my previous idea that this article would be suitable as a case report and not as a review.
We discussed the patient's illness as in a review article. That's why we claim that it is a review article.